# First Genome Description of *Providencia vermicola* Isolate Bearing NDM-1 from Blood Culture

**DOI:** 10.3390/microorganisms9081751

**Published:** 2021-08-17

**Authors:** David Lupande-Mwenebitu, Mariem Ben Khedher, Sami Khabthani, Lalaoui Rym, Marie-France Phoba, Larbi Zakaria Nabti, Octavie Lunguya-Metila, Alix Pantel, Jean-Philippe Lavigne, Jean-Marc Rolain, Seydina M. Diene

**Affiliations:** 1Faculté de Pharmacie, Aix Marseille Université, IRD, APHM, MEPHI, IHU Méditerranée Infection, 19-21 Boulevard Jean Moulin, CEDEX 05, 13385 Marseille, France; lupande2000@gmail.com (D.L.-M.); benkhedhermaryem044@gmail.com (M.B.K.); samikhabthani@outlook.fr (S.K.); l.rym-microbio@hotmail.fr (L.R.); nature_zak@hotmail.com (L.Z.N.); jean-marc.rolain@univ-amu.fr (J.-M.R.); 2Hôpital Provincial Général de Référence de Bukavu, Université Catholique de Bukavu (UCB), Bukavu 285, Democratic Republic of the Congo; 3Service de Microbiologie, Cliniques Universitaires de Kinshasa, Kinshasa 127, Democratic Republic of the Congo; mfphoba@gmail.com (M.-F.P.); octmetila@yahoo.fr (O.L.-M.); 4Département de Microbiologie, Service de Bactériologie et Recherche, Institut National de Recherche Biomédicale, Kinshasa 1197, Democratic Republic of the Congo; 5VBIC, INSERM U1047, Service de Microbiologie et Hygiène Hospitalière, CHU Nîmes, Université de Montpellier, 30000 Nîmes, France; alix.pantel@chu-nimes.fr (A.P.); jean.philippe.lavigne@chu-nimes.fr (J.-P.L.)

**Keywords:** *bla*_NDM-1_, *Providencia vermicola*, antibacterial resistance genes, NRPS/PKS cluster, comparative genomics, MDR recombinant plasmid

## Abstract

In this paper, we describe the first complete genome sequence of *Providencia vermicola* species, a clinical multidrug-resistant strain harboring the New Delhi Metallo-β-lactamase-1 (NDM-1) gene, isolated at the Kinshasa University Teaching Hospital, in Democratic Republic of the Congo. Whole genome sequencing of an imipenem-resistant clinical Gram-negative *P. vermicola* P8538 isolate was performed using MiSeq and Gridion, and then complete genome analysis, plasmid search, resistome analysis, and comparative genomics were performed. Genome assembly resulted in a circular chromosome sequence of 4,280,811-bp and 40.80% GC and a circular plasmid (pPV8538_NDM-1) of 151,684-bp and 51.93%GC, which was identified in an *Escherichia coli* P8540 strain isolated in the same hospital. Interestingly, comparative genomic analysis revealed multiple sequences acquisition within the *P. vermicola* P8538 chromosome, including three complete prophages, a siderophore biosynthesis NRPS cluster, a Type VI secretion system (T6SS), a urease gene cluster, and a complete Type-I-F CRISPR-Cas3 system. Β-lactamase genes, including *bla*_CMY-6_ and *bla*_NDM-1_, were found on the recombinant plasmid pPV8538_NDM-1, in addition to other antibiotic resistance genes such as *rmtC*, *aac6’*-*Ib3*, *aacA4*, *catA1*, *sul1*, *aac6’*-*Ib*-*cr*, *tetA*, and *tetB*. Genome comparison with *Providencia* species revealed 82.95% of average nucleotide identity (ANI), with *P. stuartii* species exhibiting 90.79% of proteome similarity. We report the first complete genome of *P. vermicola* species and for the first time the presence of the *bla*_NDM-1_ gene in this species. This work highlights the need to improve surveillance and clinical practices in DR Congo in order to reduce or prevent the spread of such resistance.

## 1. Introduction

*Providencia* species are Gram-negative bacteria belonging to the order of Enterobacterales, family of *Morganellaceae*, and genus *Providencia*. Their specific power to deaminate specific amino acids by oxidation into their corresponding keto and ammonia acids is a particularity that differentiates them from other members of the *Enterobacteriaceae* family [1,2]. Unlike other bacteria in this family, *Providencia* species are rarely involved in nosocomial infections [3]. Two species of *Providencia*, including *P. stuartii* and *P*. *rettgeri*, which are naturally resistant to many antibiotics including colistin and tigecycline, are the most common causes of more than 80% of human clinical infections, mainly urinary tract infections. The other species of this group are *P. alcalifaciens*, *P. burhodogranariea*, *P. heimbachae*, *P. rettgeri*, *P. rustigianii*, *P. sneebia*, *P. stuartii*, *P. thailandensis*, *P. huaxiensis*, and *P. vermicola* [4,5,6].

*P. vermicola* was first isolated from a nematode *Steinernema thermophilum* collected in soils in India in 2006. Its name means (ver.mi′co.la. L. n. worm; L. suff. -cola of L. n. incola inhabitant; N.L. n. *vermicola* inhabitant of worms) [7] and is very rarely found as an etiological agent in humans, with only one described case of diarrhea in a 37-year-old patient in India [8]. *Providencia* species are reported to be found mainly in environments such as water and are mostly involved in infections of birds, fish, and certain insects such as fruit flies [3,9]. Prior to this study, there was no available genome of *P. vermicola* in the NCBI database. Here, we describe the first complete genome sequence of a clinical multi drug resistant (MDR) *P. vermicola* isolate from a healthcare facility in Kinshasa, in the Democratic Republic of the Congo, and perform the comparative genomic analysis with the most closely-related species.

## 2. Materials and Methods

### 2.1. Bacterial Isolation

Two Gram-negative strains, namely *P. vermicola* P8538 and *Escherichia coli* P8540, were isolated in 2017 from blood and urine, respectively, at the KUTH (Kinshasa university teaching hospital) in DR Congo. The KUTH is a government-funded academic tertiary referral hospital in Kinshasa, the capital city of DR Congo. It is the national referral hospital in a country of approximately ninety million people. It has 1000 total beds, of which seven beds are in the intensive care unit (ICU). All isolates were identified first using biochemical tests such as urea, indole, oxidase, citrate, and triple sugar iron (TSI), and were confirmed by Microflex LT MALDI-TOF mass spectrometer (Bruker Daltonics, Bremen, Germany) after being sent to the IHU-Méditerranée infection, Marseille.

### 2.2. Antimicrobial Susceptibility Testing

Susceptibility to 16 antimicrobial agents (i.e., amikacin, amoxycillin, amoxycillin clavulanate, cefepime, ceftriaxone, cephalothin, colistin sulfate, ciprofloxacin, cotrimoxazole, doxycycline, ertapenem, fosfomycin, gentamicin, imipenem, nitrofurantoin, tazobactam piperacillin; i2a, Montpellier, France) was determined using the disk diffusion method according to the European Committee on Antimicrobial Susceptibility Testing (EUCAST) guidelines (https://www.eucast.org/clinical_breakpoints/, access on 10 February 2021). The minimal inhibition concentration (MIC) of imipenem and colistin were respectively determined by agar dilution in Mueller–Hinton agar (Oxoid, Basingstoke, UK) and by microdilution method using UMIC Colistin kit (Biocentric, Bandol, France) according to CLSI recommendations. This MIC for imipenem was determined after 18–24 h of incubation on Mueller-Hinton plates inoculated with suspensions of isolates at a fixed density (0.5 to 0.6 McFarland standard), using E-test strips (BioMérieux, Marcy l’Etoile, France) according to the manufacturer’s recommendations.

### 2.3. Molecular Mechanisms of Antibiotic Resistance and Whole Genome Sequencing

Real-time PCR and standard PCR were performed to screen for the presence of carbapenem resistance genes including: *bla*_NDM_, *bla*_OXA-23_, *bla*_VIM_, *bla*_OXA-48_, and *bla*_KPC_. Genomic DNAs (gDNA) of the two carbapenem resistant isolates were extracted using the EZ1 Advanced XL Biorobot and the tissue DNA kit (Qiagen, Hilden, Germany) with the Bacterial card, according to the manufacturer’s instructions and quantified using NanoDrop 2000 (ThermoFischer, Illkirch, France). Whole genome sequencing was performed using the MiSeq sequencer (Illumina, San Diego, CA, USA) and the Gridion sequencer (Nanopore, Oxford, UK), according to the Nanopore Template Preparation. 

### 2.4. Bioinformatic Analysis

The genome assemblies were performed using the A5-pipeline on the Illumina reads and using Unicycler for the hybrid assembly, which includes both Illumina and nanopore reads [10,11]. Genome annotation was performed using the prokka (rapid prokaryotic genome annotation) pipeline [12]. Circular genome representations of *P. vermicola* chromosome and plasmid and their comparisons by BlastN with the closest sequences were performed using the CGview Server [13] and locally downloaded EasyFig v2.2 software. Proteome comparison of *P. vermicola* with those *Providencia* species was performed using the “get_homologues.pl” pipeline [14]. Moreover, the average nucleotide identity (ANI) between *P. vermicola* and downloaded genomes was determined using the OrthoANI program (AOT software) [15]. All genes deemed to be candidates of antimicrobial resistance or putative virulence genes were investigated using the ARG-ANNOT database and VFDB with threshold value amino acid alignment ≥70% of the input query sequence to avoid any data extrapolation. PHASTER (PHAge Search Tool Enhanced Release) was used to identify prophage sequences from genomic sequences [16,17,18].

### 2.5. Conjugation Experiment

To determine the transferability of carbapenem resistance, a conjugation experiment was performed using *E. coli* J53 (azide-resistant) as the recipient strain, as previously described [19].

## 3. Results

### 3.1. Clinical Information and Phenotypic Characterisation of Isolates

The *P. vermicola* P8538 strain was isolated from the blood culture of a 58-year-old patient hospitalized in the ICU for sepsis, whereas the *E. coli* P8540 strain was isolated in the urine of a 26-year-old patient hospitalized in the same ICU, but not at the same time, and who was receiving continuous respiratory assistance. As presented in Table 1, the two isolates were resistant to the majority of the 16 tested antibiotics, with the exception of cefepime, fosfomycin, and cotrimoxazole for the *P. vermicola* P8538 isolate, whereas the *E. coli* P8540 isolate remained susceptible to colistin (Table 1).

### 3.2. Genome Features

For *P. vermicola* P8538, the genome size was estimated to be 4,432,495-bp and composed of one chromosome 4,280,811-bp in size and 40.80% GC and one plasmid (pPV8538_NDM-1) 151,684-bp in size and 51.93% GC. Genome annotation revealed 4166 predicted genes composed of 3991 CDS, 11 rRNAs, 74 tRNAs, 89 miscRNAs, and 1 tmRNA gene (Table 2).

As shown in Figure 1, genome comparison with 10 published *Providencia* genomes revealed multiple mobile genetic elements (MGEs) in the *P. vermicola* P8538 chromosome. Indeed, three complete prophage regions were identified from the PHASTER database, including Phage_Entero_mEp460 “54.42-Kb, 39.96% GC”, Phage_Salmo_Fels_2 “59.9-Kb, 41.21%GC”, and Phage_Escher_HK639 “41.7-Kb, 42.31%GC”, respectively. Additionally, a complete siderophore biosynthesis NRPS (39.62-Kb, 33.92%GC) composed of 19 genes can be identified (Figure 1). As presented in Appendix A, this siderophore NRPS was only identified from the NCBI database in three genomes including *P. stuartii* PRV00010, *Morganella morganii* VGH116, and *Salmonella enterica* 2014AM-3158 with average % homology of 96%, 44.37%, and 33.89%, respectively (Appendix A). The identified Type-I-F CRISPR-cas3 system (10.9-Kb, 43.63% GC) appeared to be specific to the *P. vermicola* P8538 isolate since it was not identified in any of the 10 compared genomes (Figure 1). Interestingly, a complete Type VI secretion system (T6SS) encoding gene was harbored in the *P. vermicola* P8538 chromosome and was partially detected in the compared genomes (Figure 1).

However, in addition to the MGEs identified from the *P. vermicola* P8538 chromosome, a complete and recombinant plasmid harboring the New Delhi metallo-β-lactamase-1 (*bla*_NDM-1_) gene has been identified from the genome sequences. As presented in Figure 2, this plasmid was mainly characterized by the presence of multiple plasmid conjugative transfer genes (14 *tra* genes), a toxin/antitoxin *higAB* system, a transposon-containing-NDM-1, a glutathione detoxification system, and four other antibiotic resistance genes. 

### 3.3. Genome Comparison with Closely Related Species

As shown in Figure 3, the whole-proteome-based phylogenetic tree and pairwise comparison of *P. vermicola* P8538 with 10 other *Providencia* species revealed that our *P. vermicola* was more closely related to *P. stuartii* genomes than those of the two recently published *P. vermicola* genomes. It appeared clear that the published *P. vermicola* G1 was wrongly identified and should be reidentified as *P. rettgeri* (Figure 3). Indeed, *P. vermicola* P8538 shared between 90.79% and 97.61% of proteome homology with *P. stuartii* and only 88.6% homology with *P. vermicola* LLDRA6 (Figure 3). 

This result was also confirmed based on the RpoB % aa identity which was 99.11% with that of *P. stuartii*, 98.96% *P. vermicola* LLDR26, and only 98.29% with *P. vermicola* G1 (Figure 3).

In addition to this evidence, as shown on Figure 4, the OrthoANI analysis revealed 82.95% and 82.89% average nucleotide identity of P8538 genome with *P. stuartii* AR_0026 and *P. stuartii* MRSN2154 genomes, respectively. Interestingly, *P. vermicola* P8538 exhibited only 81% of ANI with published *P. vermicola* LLDR26 and 77.24% of ANI with *P. vermicola* G1. These results clearly suggested a wrong identification of the latter. Indeed, *P. vermicola* G1 showed 99.24% of ANI with *P. rettgeri* 151. 

### 3.4. Resistome

Regarding the antibiotic susceptibility phenotype of the *P. vermicola* P8538 isolate (Table 1), the resistome analysis confirmed the observed β-lactam resistance phenotype by the presence of the *bla*_CMY-6,_ and *bla*_NDM-1_ genes from the plasmid pPV8538_NDM-1. Moreover, other antibiotic resistance genes were found, including *rmtC* and *aacA*4 conferring resistance to aminoglycosides, *catA*1 for phenicol resistance, *sul*1 and *aac6’-Ib-cr* for resistance to sulfonamides and quinolones, respectively (Table 1). Interestingly, the Tn3-NDM-1 transposon (21,774-bp, 59.40% GC) on the plasmid was the vehicle of four resistance genes, namely *bla*_NDM-1_, *rmtC*, *Sul*1, and *aacA*4 (Figure 2). A total of 20 virulence associated genes from the *P. vermicola* P8538 were identified and are presented in Appendix A.

### 3.5. Genomic Analysis of the E. coli P8540 Isolate

Interestingly, during this study, our investigation of the potential spread of *bla*_NDM-1_ in this hospital revealed a positive *E. coli* P8540 strain isolated from the urine of a 26-year-old hospitalized patient. This strain was subjected to whole genome sequencing and resulted in a genome assembled into 210 contigs with a size of 4,809,673-bp and 50.9% GC content. The details of the genome features are presented in Table 2. MLTS analysis reveals a type ST1412 clone and plasmid finder analysis detected five plasmid replicons which were classified as col (MG828), IncA/A2, IncFIA/(HI1), IncI, and IncR.

Interestingly, the same pPV8538_NDM-1 plasmid identified in the *P. vermicola* P8538 isolate was detected and identified from this *E. coli* P8540 isolate (pEC8540), suggesting a conjugation transfer event of this plasmid between these two *Enterobacteriaceae* species (Figure 2). Unfortunately, our in vitro experiment to transfer by conjugation the pPV8538_NDM-1 plasmid into the *E. coli* J53 (azide-resistant) strain was unsuccessful after three repeated assays.

Resistome analysis of *E. coli* P8540 confirmed the presence of the *bla*_NDM-1_, *bla*_TEM-1B_, *bla*_SHV-12_, and *bla*_CTX-M-88_ genes, conferring the observed β-lactam resistance of this isolate. Moreover, the presence of multiple genes conferring resistance to aminoglycosides (n = 12), phenicols (n = 4), sulfamide/trimethoprim (n = 2), macrolides (n = 1), quinolones (n = 18), and tetracyclines (n = 8) were identified (Table 2).

## 4. Discussion

In this paper we report the first complete genome sequences of the *P. vermicola* species. The genome analysis and comparative genomics of this clinical MDR *P. vermicola* P8538 isolate revealed significant genomic variations compared to other *Providencia* species. This may indicate the ability of this bacterium to colonize several hostile environments, given the presence of several MGEs in the genome, which is well documented in the literature [3,20,21]. We identified a conjugative and recombinant plasmid harboring antimicrobial resistance genes which was identified in two different pathogenic Enterobacterial species (*E. coli* and *P. vermicola*) from the same hospital, suggestive of a spread of the MDR plasmid within this healthcare setting. Thus, the existence of a conjugative plasmid harboring the NDM-1 enzyme in this hospital appeared to be a serious concern for infection and prevention control measures.

Some specific MGEs have been identified in this particular *P. vermicola* P8538 isolate, including a glutathione detoxification system from the plasmid pPV8538_NDM-1 which was detected by BlastN from the NCBI database in very few bacterial plasmids from *Salmonella enterica*, *Proteus mirabilis*, *Serratia marcescens*, *Enterobacter hormaechei*, *Klebsiella quasipneumoniae*, and *michiganensis.* As reported in the literature, this system is involved in the glutathione-dependent process of formaldehyde detoxification [22,23]. Moreover, we identified from the P8538 chromosome a NRPS cluster for siderophore biosynthesis which has only been identified in three other enterobacterial genomes, namely *P. stuartii* PRV00010, *M. morganii* VGH116, and *S. enterica* 2014AM-3158, and was absent in the two genomes of *P. vermicola* recently deposited in the NCBI database. As reported, siderophore systems are low molecular weight molecules which are widespread in the bacterial and fungal world, with more than 200 biosynthetic and diverse types. They play the role of capturing, solubilizing and delivering essential Fe(III) ions in the cytoplasm [24] and are involved in the growth and development of microorganisms but also in bacterial virulence, as described in *E. coli* in urinary tract infections [25]. They are also involved in bacterial dissemination by induction of inflammation in the lungs [26,27]. The T6SS acts as a virulence factor in the majority of proteobacteria with the ability to attack eukaryotic and prokaryotic target cells through a complex process, secreting toxic effectors through a contact mechanism into neighboring bacteria or eukaryotic cells, causing cell lysis or growth arrest [28,29]. This complex process involves the transport of proteins through a contractile bacteriophage-like tail structure [30,31]. Six secretion systems have thus far been identified and are referred to as Type I to Type VI (T1SS to T6SS). T6SS was first discovered in *Vibrio cholerae* and *Pseudomonas aeruginosa* in 2006, and several studies have subsequently demonstrated its presence in many Gram-negative bacteria, including many human and animal pathogenic strains [32]. Indeed, studies have shown that certain T6SS subunits have structural homologies with other subunits of the bacteriophage T4, including the main tail protein and its injection syringe. It has therefore been established that T6SS is phylogenetically and structurally very close to the bacteriophage T4 [32]. Thus, we believe that the integration of several exogenous sequences, including bacteriophages and T6SS, may play a role in the adaptation and survival of *P. vermicola* which evolves in endosymbiosis in *Steinernema thermophilum*, a nematode in which *P. vermicola* develops [33].

## 5. Conclusions

This study highlights the emerging threat of *bla*_NDM-1_ dissemination in Kinshasa. To the best of our knowledge, this study describes, for the first time in the Democratic Republic of the Congo, the *bla*_NDM-1_ gene, in a bacterial genus of *Enterobacteriaceae* and in a rare species (*P. vermicola*), about which very little is known in Africa in general and nothing is known on the genomic level for this species. The fact that *P. vermicola* has been widely described as a nematode endosymbiont, nematodes which infect and kill fish and insects, some of which are used as food in certain environments such as Kinshasa, gives rise to speculation about the role that the habit of humans eating insects might play in the transmission of this bacteria. In addition, *S. thermophilum*, a nematode in which *P. vermicola* develops, is not recognized as a human pathogen. It cannot be excluded that this microorganism could be found in other parasites, in this case causing bacteria–parasite co-infections in humans. The identification of these NDM-1-producing isolates, which are also resistant to several other antibiotics and shared through the same plasmid with another isolate (P8540) in the same healthcare facility, confirms the existence of mobile genetic element exchanges among the circulating isolates within the University Hospital of Kinshasa. Therefore, it is urgent to improve surveillance and clinical practices to reduce or prevent the spread of resistance.

## Figures and Tables

**Figure 1 microorganisms-09-01751-f001:**
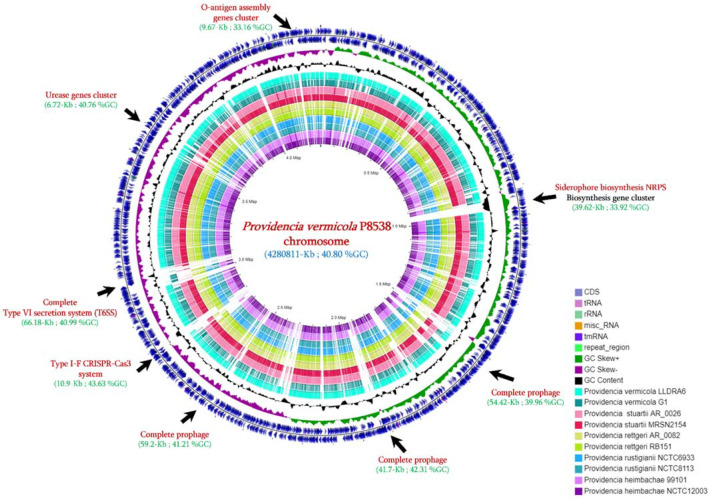
Circular map of the *P. vermicola* P8538 chromosome and its genomic comparison with the nine closest *Providencia* genomes.

**Figure 2 microorganisms-09-01751-f002:**
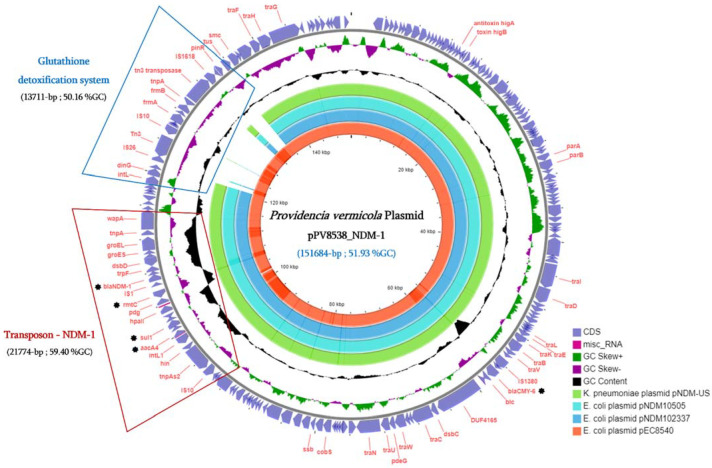
Comparison of the *P. vermicola* pPV8538_NMD-1 plasmid with *E. coli* pEC8540 plasmid and the three closest plasmids retrieved from the NCBI database.

**Figure 3 microorganisms-09-01751-f003:**
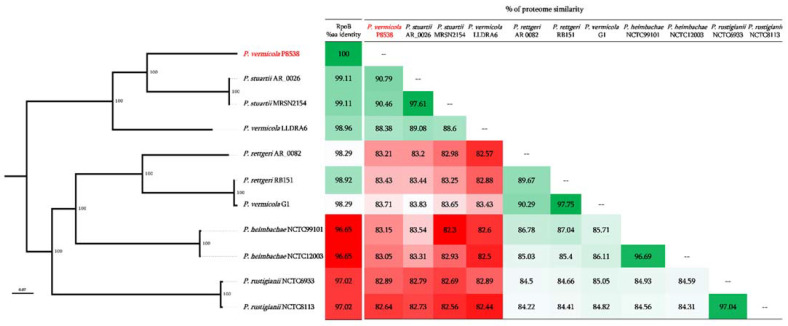
Proteome Pairwise comparison and the *RpoB* phylogenetic tree of *Providencia* genomes.

**Figure 4 microorganisms-09-01751-f004:**
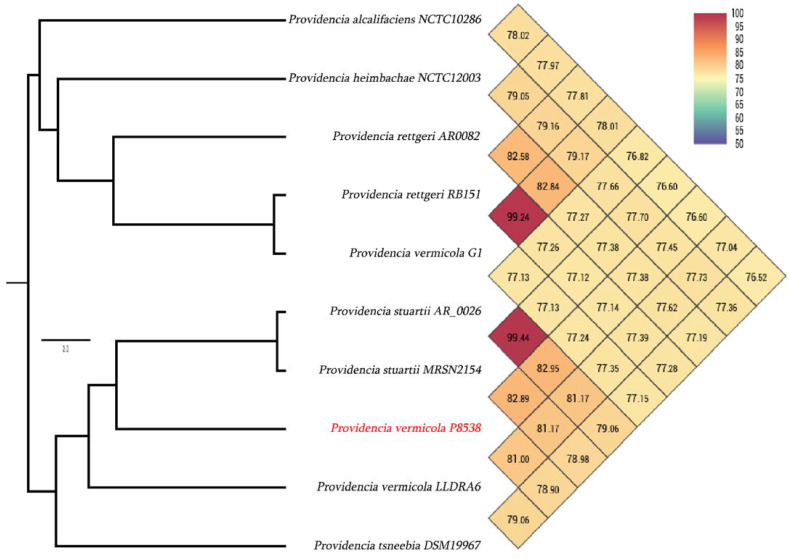
Average nucleotide identity comparison of *P. vermicola* P8538 with *Providencia* species using the OrthoANI software.

**Table 1 microorganisms-09-01751-t001:** Clinical features, Resistance phenotype, and genotype of *P. vermicola* P8538 and *E. coli* P8540 harboring the *bla*_NDM-1_ gene.

Strain Names	Sex	Age	Sample	Status	Service	Resistance Phenotype	MIC IPM (µg/mL)	Antimicrobial Resistance Genes					
								β-lactams	Aminoglycosides	Sulfonamides and Trimethoprim	Phenicols	Quinolones and Imidazoles	Cyclines
*P. vermicola* P8538	M	58	Blood	Inpatient	ICU	AX, AMC, TZP, KF, CRO, ERT, IPM, AK, GEN, CIP, DO	>32	*bla*_CMY-6_, *bla*_NDM-1_, H-NS	*rmtC*, *aac*(6′)-*Ib*3, *aac*(6′)-*Ib*10, *aac*(2)-Ia	*sul*1	*catA*1	*aac*(6′)-*Ib*-*cr*, *acrB*, *msbA*, *mdtH*, *crp*	*tetA*, *tetB*, *tetD*
*E. coli* P8540	F	26	Urine	Inpatient	ICU	AX, AMC, KF, CRO, FEP, ERT, IPM, AK, GEN, CIP, DO	>32	*bla*_TEM-1B_, *bla*_SHV-12_, *bla*_AmpC1_, *bla*_CMY-42_, *evgS*, *evgA*, *bla*_CTX-M-88_, *bla*_NDM-1_, *bla*_CTX-M-15_, *ampH H-NS*	*aph(6)-Id*, *aph(3′’)-Ib*, *aadA2*, *aadA1*, *aac(3′)-IId*, *aac(6′)-Ib3*, *rmtC*, *aadA16*, *baeR*, *baeS*, *strB*, *strA*	*dfrA*27, *dfr*12	*catA2*, *aadA2*, *mdtm*, *catII*	*qnrB6*, *qnrS1*, *aac(6′)-Ib-cr*, *qepA*, *emrR*, *emrA*, *emrB*, *mdtE*, *mdtH*, *mdtF*, *gadW*, *gadX*, *acrB*, *acrA*, *crp*, *acrE*, *acrF*, *aadA1-pm*,	*tet(D)*, *tet(A)*, *tetR*, *tetD*, *tet34*, *mdfA*, *emrK*, *emrY*

AX: Amoxycillin, AMC: Amoxicillin-clavulanate, TZP: Piperacillin tazobactam, KF: Cefazoline, CRO: Ceftriaxone, FEP: Cefepime, ERT: Ertapenem, IPM: Imipenem, AK: Amikacin; GEN: Gentamicin, CIP: Ciprofloxacin, DO: Doxycycline, SXT: Sulfamethoxazole-trimethoprim, ICU: Intensive care Unit.

**Table 2 microorganisms-09-01751-t002:** Genome features of *P. vermicola* P8538 and *E. coli* P8540 isolates harboring *bla*_NDM-1_.

Features	*P. vermicola* P8538	*E. coli* P8540
Genome size	4,432,495-bp	4,809,673-bp
% GC content	41.1%	50.9%
No. of contigs	2	210
N50	184,648-bp	80,966-bp
No. of predicted genes	4166	4951
No. of CDS	3991	4553
No. of predicted tRNAs	74	83
No. of predicted rRNA	11	13
No. of predicted tmRNA	1	1
No. of predicted miscRNAs	89	303
No. of phage sequences	5	2
Sequence Type (ST)	-	1421

## Data Availability

Not applicable.

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
