# Peer review of "First Genome Description of Providencia vermicola Isolate Bearing NDM-1 from Blood Culture"

_microorganisms, 2021, doi:10.3390/microorganisms9081751_

Round 1
Reviewer 1 Report
The paper presents the results of whole genome sequencing of Providencia vermicola isolate carrying NDM-1 carbapenemase. The work is done at a good level; however, minor shortcomings should be noted
- It is preferable to use modern taxonomy: Order: Enterobacterales; Family: Morganellaceae; Genus: Providencia
- Lines 111-113: In Methods section authors indicated that conjugation experiments were performed. However, results are not presented.
- There are inaccuracies in the antibiotic susceptibility section.
- Lines 119-121: the two isolates were resistant to the majority of the tested antibiotics, with the exception of cefepime, … However, in Table 1 coli P8540 isolate is indicated as FEP resistant.
- Lines 121-122: E. coli P8540 isolate remained susceptible to colistin. However, in Table 1 data on colistin resistance/susceptibility are not presented. Moreover, disk-diffusion is not recommended for colistin susceptibility testing by EUCAST and CLSI. No genes or mutations responsible for colistin resistance were detected in P. vermicola P8538 221 isolate.
Reviewer 2 Report
Dear authors,
I have some comments and a few questions regarding the manuscript.
Please, write DR as entire words in the abstract, the first time you use this acronym.
In the sentence "We report for the first complete genome of P. vermicola species and for the time the presence of the blaNDM-1 gene in this species. This work highlights the need to improve surveillance and clinical practices in DR Congo in order to reduce or prevent the spread of such resistance." probably the "first time" words were improperly divided.
The genome sequence of E. coli strain is not mentioned in the abstract and the introduction. Why? It is better to write Escherichia the first time L66.
The quality of all the figures should be improved and the names of the strains should be coherent along with the manuscript. Check between the figures and the main text. As an example, Figure 2. Comparison of the P. vermicola plasmid p located in P8538 with P8540 and the three closest plasmids retrieved from the NCBI database.
In the MM is reported "To determine the transferability of carbapenem resistance, a conjugation experiment was performed using E. coli J53 (azide-resistant) as the recipient strain, as previously described [19].", I have two comments, first this experiment is not present in the results, second it should be described under the paragraph Bioinformatic analysis.
The manuscript is written using in some parts the present and others the past. Please, try to be more consistent along with the text.
Round 2
Reviewer 2 Report
The authors addressed all my concerns and improved the manuscript.
However, the first E. coli in the abstract should be replaced by Escherichia coli.
Regards